# Harnessing flavonoids to control probiotic function: *in situ* application of a naringenin-responsive genetic circuit

Brenno Wendler Miranda,[1] Lucas Henrique Junges,[1] Emanuel Maltempi de Souza,[1] Paula Santana Lunardi,[2] Marcelo Müller-Santos[1]

**ABSTRACT** Efforts to improve probiotics' capabilities through synthetic biology have recently increased. Using naturally occurring plant-based compounds, such as flavonoids, as inputs of genetic circuits introduced in probiotics is a promising strategy to enhance or introduce a beneficial phenotype to the host. However, flavonoid-responsive genetic circuits have not yet been applied in host-microbe conditions. In this work, we have characterized the FdeR-based naringenin-responsive genetic circuit in the probiotic (Escherichia coli Nissle 1917) (EcN), both *in vitro* and *in situ*, applying the FdeR system in EcN colonizing mice guts. In culture, the circuit demonstrated a 131-fold activation upon naringenin exposure. In mice, circuit activity was monitored via luminescence produced by Nanoluc in fecal samples following oral gavage of naringenin (100 mg/kg), resulting in a 34-fold luminescence increase. This activation decreased over 24 hours but was reinduced after a second gavage with naringenin. These findings demonstrate the potential of naringenin-responsive genetic circuits to program probiotic phenotypes *in vivo* through external compound administration. Future studies should evaluate lower naringenin dosages and naringenin-rich foods as alternative inputs.

**IMPORTANCE** Engineering probiotics is a rapidly advancing field in synthetic biology. Genetic circuits, which enable precise and predictable control of microbial phenotypes, are central to this effort. Leveraging natural, plant-based compounds like flavonoids to control gene expression offers a promising strategy for developing next-generation probiotics with enhanced capabilities. This study demonstrates the activity of a flavonoid-responsive genetic circuit during host-microbe interactions. The findings provide a foundation for using beneficial compounds like naringenin as inputs to drive desired phenotypes *in vivo*. In addition, this work expands the range of genetic circuit inputs, facilitating the design of more sophisticated, multi-input systems for probiotic engineering.

**KEYWORDS** *E. coli* Nissle 1917, synthetic biology, probiotics, naringenin, FdeR

**Peer Reviewer** Shaohua Wang, Ohio University Heritage College of Osteopathic Medicine, Athens, Ohio, USA

Address correspondence to Marcelo Müller-Santos, marcelomuller@ufpr.br.

The authors declare no conflict of interest.

Probiotics are increasingly recognized as versatile tools for promoting host health and addressing various conditions, such as allergies, lactose intolerance, and chronic diarrhea (1–4). Despite their benefits, the functionalities of probiotics are limited by their native genetic repertoire and the environmental cues encountered in host tissues. Synthetic biology offers tools to overcome these limitations by engineering probiotics with expanded functionalities, transforming them into living therapeutics (5, 6).

Central to probiotic engineering is the design and deployment of genetic circuits, which program living organisms to perform specific tasks in response to certain input (7). While numerous small molecules have been explored as inducers, few accessible and non-toxic compounds have been successfully applied *in vivo*. Notably, compounds such as arabinogalactan and rhamnose have effectively activated genetic circuits in *Bacter-*

10.1128/spectrum.02890-24  **1**

*oides thetaiotaomicron* (8), highlighting the potential of using plant-based compounds as inputs to genetic circuits carried by living therapeutics.

Flavonoids, naturally occurring plant molecules, are promising inputs for genetic circuits in probiotic engineering due to their antioxidant, anti-inflammatory, and potential health benefits (9) while exhibiting low toxicity (e.g., naringenin has an $LD_{50}$ of 600 mg.kg$^{-1}$) (10). They are widely available in plant-based foods and are commercially used as food supplements. Thus, using a flavonoid-responsive genetic circuit in probiotics can harness the benefits of these compounds while enabling the controlled expression of a beneficial phenotype in the host.

The FdeR-PfdeA system, derived from the endophytic bacterium *Herbaspirillum seropedicae*, represents a robust flavonoid-responsive genetic circuit. When associated with naringenin, the transcriptional regulator protein FdeR activates the $P_{fdeA}$ promoter and drives the downstream genes' transcription (11). Genetic circuits utilizing this system have been successfully constructed and characterized (12–15). However, the activity of such circuits within the dynamic environment of the gut remains uncharacterized, where challenges such as flavonoid bioavailability, host-microbe interactions, and environmental variability may influence performance. In this study, we address these knowledge gaps by employing a naringenin-responsive genetic circuit in the probiotic (Escherichia coli Nissle 1917) (EcN). EcN, a widely studied and well-characterized probiotic, was chosen for its ability to colonize the gut and its amenability to genetic manipulation. We evaluated the circuit's activation both *in vitro* and *in situ*, using naringenin as an input to drive reporter gene expression in bacteria colonizing the mouse gut. By comparing the circuit's performance in these conditions, we provide new insights into the potential and limitations of flavonoid-responsive genetic circuits for programming probiotic phenotypes *in vivo*.

*E. coli* Top10 was used for plasmid construction and maintenance, while *E. coli* (Nissle 1917) (EcN) was used for circuit characterization. The strain EcN *attλ::sfgfp* was generated as previously described (16) by inserting a DNA cassette for constitutive GFP expression at the *attλ* site. Transformants were selected on LB agar containing 50 µg/mL kanamycin, and the helper plasmid was cured by growing the cells at 42°C. All cultures were grown in LB medium or LB agar at 37°C with shaking at 180 rpm. For full experimental details, please refer to the Supplemental Material.

EcN strains carrying the plasmids 114-*fdeR-sfgfp* or 114-*fdeR-nanoluc* were cultured, and fluorescence or luminescence was measured after adding naringenin (up to 100 µM). GFP fluorescence was monitored over 15 hours using the Tecan Infinite 200 PRO plate reader, while Nanoluc-based luminescence was measured after 4 hours using the Synergy LX plate reader. Due to challenges in measuring sfGFP fluorescence in fecal samples, such as interference from fibers and endogenous fluorescent compounds, we used the NanoLuc-based luminescence system to characterize the genetic circuit's activity *in situ*.

*In situ* experiments were conducted with 10 C57BL/6 mice randomly assigned to two groups (three males and two females per group): one group received 100 mg/kg of naringenin (NAR group) in a 0.5% carboxymethylcellulose suspension. By contrast, the control group (CMC group) received an equivalent volume of 0.5% carboxymethylcellulose solution based on the animals' body weight. On days −3 and −2 prior to the administration of EcN λ::*sfgfp* transformed with 114-*fdeR-nanoluc* (hereafter EcN/114-*fdeR-nanoluc*), each animal was treated with 20 mg of streptomycin. On day −1, 1 g/L ampicillin was added to the drinking water, which was refreshed every 24 hours and maintained until the end of the experiment. On day 0, all mice were gavaged with $10^9$ CFU/mL of EcN/114-*fdeR-nanoluc*. On days 2 and 3, the mice received either CMC or NAR via gavage. From day 1 onward, feces were collected daily, weighed, and homogenized in 1 mL of PBS buffer. After treatment, feces were collected 2-, 6-, and 24 hours post-gavage (Fig. 2A). A 10 µL aliquot of each sample homogenate was used for NanoLuc assays (Promega, USA), while the remaining material was processed for DNA extraction to quantify plasmid concentrations with qPCR, which was used to normalize luminescence

data (see Supplemental Material for details). Plasmid quantitation by qPCR is preferred over CFU counting for luminescence normalization as it provides a more accurate and consistent measure of plasmid abundance, minimizing variability caused by differences in bacterial viability and extraction from fecal samples (17). The values obtained were analyzed by the Mann-Whitney test, with statistical significance considered as $P < 0.05$.

We first constructed and characterized the 114-*fdeR-sfgfp* circuit in *E. coli* (Nissle 1917) (EcN) under *in vitro* conditions. This circuit exhibited a 113-fold fluorescence increase 4 hours after adding 100 µM naringenin (Fig. 1A), consistent with previously reported data for *E. coli* MG1655 (12). To better simulate the timing of gene expression in the murine gut, we also monitored *sfgfp* expression over 15 hours following induction. Although the activation rate slowed over time, the dynamic range—measured as the difference in specific fluorescence between 100 µM and 0 µM naringenin—significantly increased after 15 hours (Fig. S1).

For *in situ* characterization, we replaced the reporter gene with *nanoluc* (114-*fdeR-nanoluc*), which encodes a highly sensitive 19 kDa luciferase, following a strategy similar to a previous study (8) and chosen for its ability to overcome fluorescent measurement challenges in complex environments. The Nanoluc activity increased 11-fold upon naringenin addition (Fig. 1B), confirming the circuit's responsiveness. The construction was transformed into EcN (λ::*sfgfp*) to assess the activity of the 114-*fdeR-nanoluc* in *E. coli* colonizing the murine gut. This strain effectively colonized (Fig. S3) and activated *nanoluc* expression in the murine gut, as evidenced by the consistently higher median luminescence values in the NAR group compared to the CMC group throughout the treatment period (Fig. 2B). Significant differences in luminescence were observed between the groups at all time points post-naringenin administration, except for the 2 h mark after the first gavage. Notably, the circuit's peak activation occurred 6 h after the first dose of naringenin, with luminescence levels in the NAR group being 34 times higher than in the CMC group.

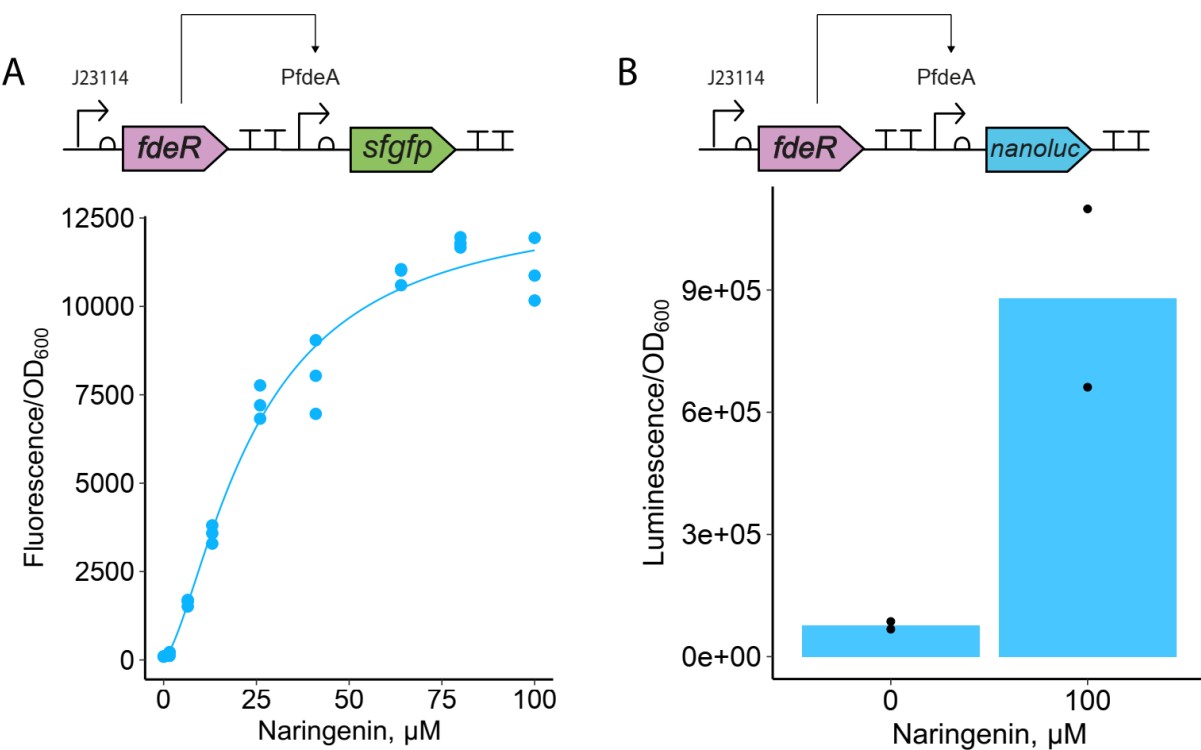

**FIG 1** *In vitro* characterization of FdeR-dependent genetic circuits in EcN. (A) *sfgfp* was used as the reporter gene to describe the genetic circuit's kinetics. The curve represents a fit to the Hill equation, and each point represents individual measures ($n$ = 3). (B) Luminescence measurement for the strain EcN/114-*fdeR-nanoluc* induced with 100 µM naringenin. Data points represent the luminescence values normalized by optical density ($OD_{600}$).

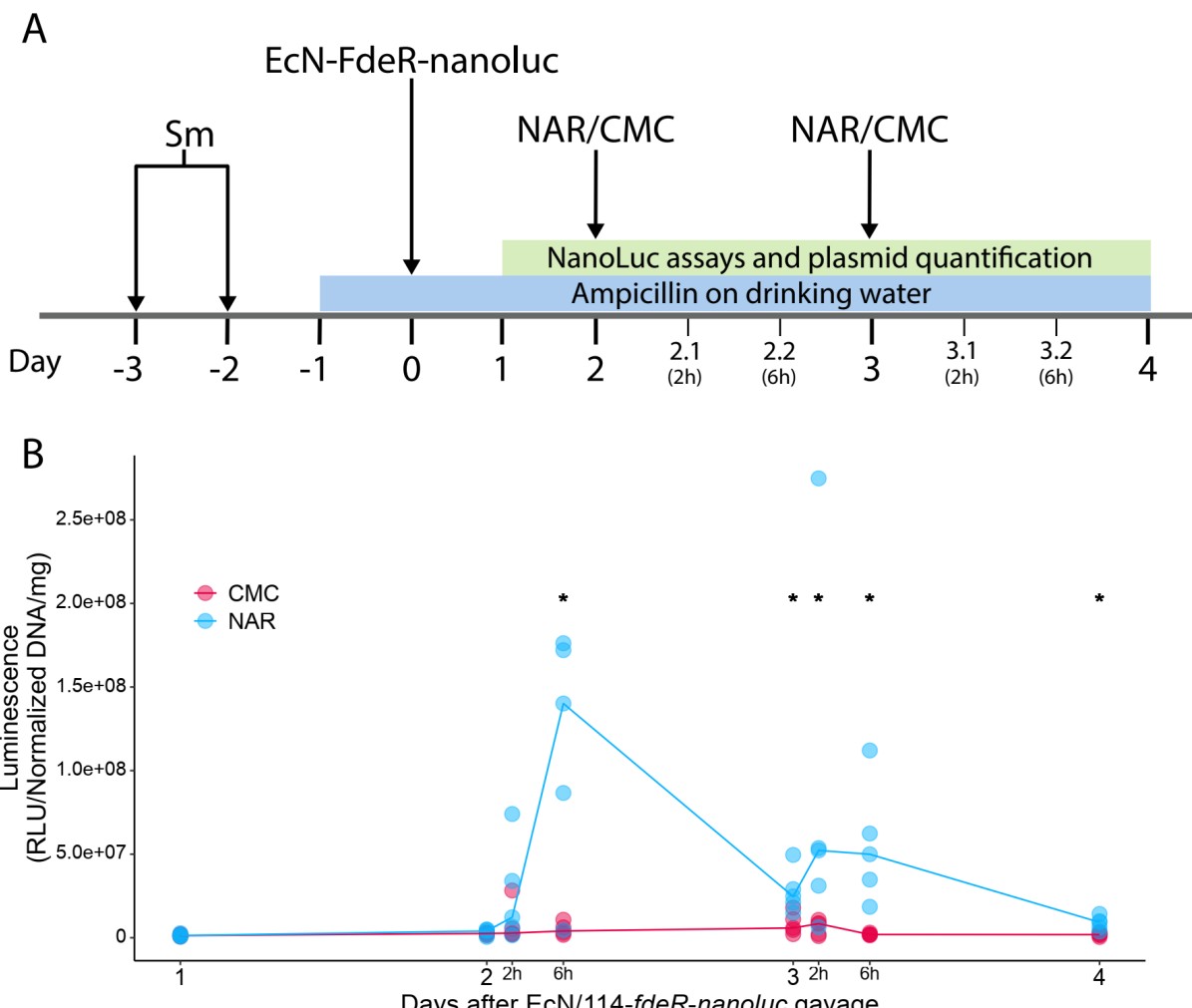

**FIG 2** *In vivo* characterization of FdeR genetic circuits. (A) Experimental timeline. Arrows indicate treatments administered by gavage. Days with decimal points indicate samples collected 2 and 6 hours after NAR/CMC gavage and are not represented on scale. (B) Luminescence from mice feces, expressed as relative luminescence units normalized by plasmid abundance ([plasmid DNA]/[total DNA]) and fecal mass. Points represent individual samples, with lines connecting each group's median values. Asterisks indicate statistically significant differences ($P < 0.05$) between NAR and CMC, determined using the Mann-Whitney test. CMC: carboxymethylcellulose; NAR: naringenin; Sm: Streptomycin.

The temporal expression pattern is also noteworthy, with a peak in the first 6 h and a decline to near-control levels after 24 h. This pattern repeated after the second NAR gavage, although peak luminescence values were lower than those observed after the first dose. Likely, the emergence of an EcN population with reduced or silenced *nanoluc* expression has contributed to the luminescence reduction after the second gavage with naringenin, a possibility that needs to be addressed in further work.

An apparent temporal effect on reporter gene expression was observed, with activity declining 24 h post-gavage. This decrease likely reflects naringenin's metabolism, transport, and elimination, which diminish significantly over this period (18, 19). Although we considered alternative methods for administering the inducer, like providing it *ad libitum* in drinking water, the low solubility of naringenin in water made this approach impractical. The *in vivo* expression profile demonstrated here is attractive for applications requiring transient phenotypic responses, such as enzyme production after a meal.

In this study, we successfully engineered and characterized a naringenin-responsive genetic circuit in EcN, demonstrating its functionality both *in vitro* and *in situ*. Our

findings highlight the potential of using flavonoids, specifically naringenin, as an external input to control gene expression in probiotics colonizing the murine gut.

The *in vitro* characterization of the 114-*fdeR-sfgfp* circuit revealed a 113-fold fluorescence increase upon adding 100 µM naringenin, corroborating previous work in *E. coli* MG1655 (12). This result establishes naringenin as a potent and reliable inducer of the FdeR-P$_{fdeA}$ system in *E. coli*. Interestingly, when gene expression was tracked over 15 hours, the dynamic range increased despite a reduced activation rate, indicating that the system remains active over extended periods. This extended activation profile could be advantageous in applications requiring prolonged gene expression, such as the sustained production of therapeutic proteins.

When applied *in situ*, our genetic circuit continued to show robust activity in the murine gut. The EcN/114-*fdeR-nanoluc* strain displayed significant luciferase activity following naringenin administration, with luminescence levels peaking 6 hours post-gavage and declining to near baseline levels by 24 hours. This temporal expression pattern is consistent with the known pharmacokinetics of naringenin, where rapid metabolism and clearance likely contribute to the reduction in gene expression over time (18). These findings suggest that flavonoid-responsive genetic circuits may be particularly suited for applications that require transient expression, such as time-specific responses linked to host metabolism or environmental cues.

One of the most striking observations was the 34-fold increase in luminescence in the naringenin-treated group compared to the control (CMC) group 6 hours after the first gavage. This marked circuit activation demonstrates the high sensitivity and dynamic range of the FdeR-PfdeA system in response to naringenin in a complex environment. Such high sensitivity is particularly promising for therapeutic applications, where precise control over gene expression is critical. In addition, the ability to induce gene expression in the gut through a safe, plant-derived compound like naringenin opens new avenues for non-invasive modulation of gut microbiota.

However, several points must be considered before optimizing this system. First, the high dose of naringenin used in this study (100 mg/kg) was selected for its established therapeutic relevance (10). A naringenin dosage of 100 mg/kg to mice translates to approximately 2 mg for a 20 g mouse. To contextualize this dosage regarding human dietary intake, it is important to consider the naringenin content in common citrus fruits, such as grapefruits, achieving naringenin concentrations ranging from 62 to 308 mg/kg (20). Considering these values, achieving a naringenin intake of 100 mg/kg in humans through diet alone would require consuming an impractically large quantity of citrus fruits. Therefore, while dietary sources can contribute to naringenin intake, they are unlikely to reach the concentrations used in our experimental setup. For therapeutic applications aiming to replicate the effects observed in our study, naringenin supplementation or pharmacological formulations containing the EcN strain and an activator dose of naringenin may be more practical and effective than relying solely on dietary intake.

While a 100 mg/kg dose effectively induces gene expression, future studies could explore lower concentrations to determine the minimal effective dose and assess whether gene expression can be fine-tuned by adjusting the input concentration. Strategies to amplify the input concentration of naringenin in the EcN colonizing gut, introducing modules expressing T7 RNA polymerase or quorum sensing genes, could also be explored in future works. Second, naringenin's low water solubility limited our administration options, preventing continuous delivery methods such as *ad libitum* drinking water supplementation. Developing more soluble formulations of naringenin or other flavonoids could improve the practicality of sustained induction over longer periods, potentially broadening the range of applications for this system.

In addition, we employed an antibiotic-treated mouse model to evaluate the functionality of a naringenin-responsive genetic circuit in the gut. While this model has limitations regarding its direct applicability to human microbiota (21), it serves as a valuable first step for demonstrating the potential of flavonoids as inducers of gene

expression *in vivo*. By reducing the complexity of the gut environment, this approach allows for a clearer assessment of the genetic circuit's performance without interference from native microbial populations. The controlled conditions ensure that observed changes in luminescence can be attributed to the activation of the genetic circuit by naringenin rather than confounding interactions with the existing microbiota. Besides, future work could explore alternative strategies for maintaining plasmid selection that do not rely on antibiotics. One promising approach is auxotrophic markers, where the engineered strain is complemented with a plasmid carrying a gene essential for survival in a defined medium (22). This strategy eliminates the selective pressure exerted by antibiotics and reduces the disruption of the natural gut microbiota, making the system more applicable to real-world scenarios. In addition, auxotrophic selection can provide a more stable and environmentally friendly method for plasmid maintenance, facilitating long-term studies and potential therapeutic applications in more natural microbiota contexts. Such efforts would enhance the translational potential of flavonoid-responsive genetic circuits for use in probiotics and other living therapeutics.

Our work represents the first report of using flavonoids as inputs for programming probiotics, expanding the repertoire of natural compounds employed in synthetic biology circuits. Given the growing interest in plant-based therapeutics and their generally favorable safety profiles, flavonoids may represent an ideal class of molecules for future development in probiotic-based therapies. Moreover, leveraging the beneficial properties of flavonoids—such as their antioxidant and anti-inflammatory activities—could enable the simultaneous delivery of therapeutic compounds and the induction of beneficial phenotypes in the gut microbiome.

In conclusion, this study demonstrates the feasibility of using a flavonoid-responsive genetic circuit to control gene expression in both *in vitro* and *in situ* probiotics. The results provide a cornerstone for future research to refine and expand this system for therapeutic and diagnostic applications. Further exploration of other plant-derived molecules and broader induction conditions will likely yield additional opportunities for advancing the field of living therapeutics.

## ACKNOWLEDGMENTS

The authors gratefully acknowledge the financial support provided by CAPES (Coordenação de Aperfeiçoamento de Pessoal de Nível Superior) and CNPq (Conselho Nacional de Desenvolvimento Científico e Tecnológico) for funding this research. We sincerely thank Dr. Ana Claudia Bonatto, Dr. Edileusa Gerhardt, and Dr. Fernanda Kashiwagi for their critical analysis and contributions to the manuscript. In addition, we appreciate the technical support provided by Roseli Prado, Valter Baura, and Alex Tramontim, whose assistance was essential to accomplish this work.

## AUTHOR AFFILIATIONS

[1]Postgraduate Program in Science (Biochemistry), Department of Biochemistry and Molecular Biology, Nitrogen Fixation Laboratory, , Federal University of Paraná (UFPR), Curitiba, Brazil
[2]Postgraduate Program in Science (Biochemistry), Department of Biochemistry and Molecular Biology, Biological Oxidations Laboratory, , Federal University of Paraná (UFPR), Curitiba, Brazil

## AUTHOR ORCIDs

Marcelo Müller-Santos  http://orcid.org/0000-0001-9615-6007

## AUTHOR CONTRIBUTIONS

Brenno Wendler Miranda, Conceptualization, Data curation, Formal analysis, Investigation, Methodology, Writing – original draft, Writing – review and editing | Lucas Henrique Junges, Data curation, Formal analysis, Investigation, Methodology, Writing – original

draft, Writing – review and editing | Emanuel Maltempi de Souza, Conceptualization, Data curation, Formal analysis, Validation, Writing – original draft, Writing – review and editing | Paula Santana Lunardi, Conceptualization, Data curation, Formal analysis, Methodology, Supervision, Validation, Writing – review and editing | Marcelo Müller-Santos, Conceptualization, Data curation, Formal analysis, Funding acquisition, Investigation, Methodology, Project administration, Resources, Software, Supervision, Validation, Writing – original draft, Writing – review and editing

## ADDITIONAL FILES

The following material is available online.

## Supplemental Material

**Supplemental material (Spectrum02890-24-s0001.docx).** Detailed methodologies, additional figures, and supporting data for the characterization of the naringenin-responsive genetic circuit in probiotic *E. coli* Nissle 1917, both in vitro and in vivo.

## Open Peer Review

**PEER REVIEW HISTORY (review-history.pdf).** An accounting of the reviewer comments and feedback.

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
