## [Reviewer comments · Microbiology Spectrum]

Microbiology Spectrum

Harnessing Flavonoids to Control Probiotic Function: In Situ Application of a Naringenin-Responsive Genetic Circuit

Brenno Wendler Miranda, Lucas Junges, Emanuel Souza, Paula Santana Lunardi, and Marcelo Müller-Santos

Corresponding Author(s): Marcelo Müller-Santos, Federal State University of Paraná

Review Timeline:

Submission Date:	November 11, 2024
Editorial Decision:	December 11, 2024
Revision Received:	January 24, 2025
Accepted:	February 22, 2025

Editor: Ilana Kolodkin-Gal

Reviewer(s): Disclosure of reviewer identity is with reference to reviewer comments included in decision letter(s). The following individuals involved in review of your submission have agreed to reveal their identity: Shaohua Wang (Reviewer #1)

Transaction Report:

DOI: <https://doi.org/10.1128/spectrum.02890-24>

Re: Spectrum02890-24 (Harnessing Flavonoids to Control Probiotic Function: In Situ Application of a Naringenin-Responsive Genetic Circuit)

Dear Prof. Marcelo Müller-Santos:

Thank you for the privilege of reviewing your work. Below you will find my comments, instructions from the Spectrum editorial office, and the reviewer comments.

Revision Guidelines

Sincerely,
Ilana Kolodkin-Gal
Editor
Microbiology Spectrum

Reviewer #1 (Comments for the Author):

In this manuscript, authors generated flavonoids (naringenin) inducible gene expression system, which was transformed into a probiotic E. Coli strain to control functional gene expression. Both in vitro and in vivo results demonstrated the induced protein expression. Because the authors have published a very similar work for developing the naringenin-induced protein expression system, the in vitro work from this manuscript has no novelty. It is important to explore the feasibility of applications of a

naringenin-responsive genetic circuit, but lots of defects were noticed.

1. They described that EcN colonized the mice gut, but there are no experiments performed, and no data verified its colonization.
2. Similar with the above, the EcN strain has chromosomal inserted gfp gene, but expression of GFP was not detected which may indicate the growth and colonization of this strain.
3. The in situ mice model used applied streptomycin treatment, I thought the purpose was to disturb the normal gut microbiome, but the authors did not discuss any details related to this like the limitations of this model for further translation applications.
4. Ampicillin was also used during the treatment to keep the plasmid, but at the same time ampicillin will also eliminate lots of gut microbes in the gut, making it not a real normal gut condition. As it has been done for inserting gfp gene into the chromosome, it is better to integrate the target gene to the genome of the probiotic strain.
5. No discuss about the dose use (100 mg/kg of naringenin) is comparable to that from the naringenin rich food?
6. No clear discussion about the weak peak came after the second induction compared to the first gavage treatment.
7. In Figure 2B, time is labeled as days, it is better to label the 2 h and 6 h because the results were described as hours.
8. In Table S1, some of the strains used/generated in this study were not listed

Reviewer #2 (Comments for the Author):

The study entitled "Harnessing flavonoids to control probiotic function: in situ application of a naringenin-responsive genetic circuit" lays a strong foundation for the use of flavonoid-responsive genetic circuits in probiotic applications. However, challenges related to dose optimization, solubility, and interindividual variability must be addressed to unlock the full potential of this system. Future research should focus on integrating these circuits into more complex therapeutic strategies, including combination therapies and feedback-controlled systems, to create next-generation living medicines. The authors could enhance the clarity and impact of their work, following the below suggestions.

1. Abstract

Revise the abstract to remove errors, avoid redundancy, and streamline the presentation of results and conclusions. e.g. repeated phrases such as "as it presented minimal levels of basal output" occur, which disrupts the flow of information. Typos such as "probioitics" and "timetime". The authors are encouraged to conduct thorough proofreading to eliminate typographical errors, awkward phrasing, and formatting issues.

2. Importance

- * The introduction spends excessive space reiterating basic concepts about probiotics, synthetic biology, and flavonoids, which may be redundant for the target audience of a specialized publication. Example, "certain microbes are probiotics, offering health benefits to their hosts when taken in appropriate amounts" is unnecessary for readers familiar with the field.
- * The introduction does not sufficiently highlight the knowledge gaps addressed by the study, such as the challenges of achieving consistent genetic circuit activation in complex gut environments. Example, a more direct comparison between the performance of genetic circuits in vitro and in situ could frame the novelty of the work more clearly.
- * Expand the discussion on why flavonoids are particularly promising inputs by addressing their gut stability, metabolic pathways, and interactions with microbial systems.
- * Balance the level of detail across sections to create a cohesive narrative. For instance, provide a brief explanation of why E. coli Nissle 1917 was chosen, alongside details about the FdeR-PfdeA system.

3. Materials & methods

- * Streamline the text by eliminating repeated phrases and consolidating similar information. For instance, combine the fluorescence and luminescence measurement descriptions into one cohesive statement.
- * Include specific details such as the naringenin concentration range, plate reader settings, and qPCR cycling conditions in the main text to enhance reproducibility.
- * Explain why GFP and Nanoluc reporters were chosen and how their use contributes to the study's goals. For instance, discuss the sensitivity or stability advantages of these reporters in gut environments.
- * Add critical details about the animal model, including the number of mice, grouping strategies, any pre-treatments like antibiotics, and ethical compliance.
- * While supplementary material can contain extensive protocols, ensure that the main text provides sufficient standalone information for readers to grasp the experimental setup.

4. Results

- * While the study uses 100 μ M naringenin and 100 mg/kg doses, it does not explore the full dose-response relationship to assess threshold levels or optimize induction conditions. Example, no mention is made of how lower or higher doses influence the circuit, limiting the understanding of its sensitivity and range of application.
- * While the temporal decline in gene expression is acknowledged, the study does not explore how repeated dosing or alternative administration strategies could sustain circuit activity. Example, administering naringenin via food or developing water-soluble derivatives might overcome the solubility issue and ensure prolonged activation.

* The results rely on luminescence normalized to plasmid DNA concentration, but there is no discussion of variability in plasmid retention or expression across the bacterial population in the gut. Example, plasmid stability under gut conditions could vary, potentially influencing circuit performance.

* Although differences between groups are described as significant, the methods for determining significance are not detailed, leaving readers uncertain about the robustness of these claims. Example, specific statistical tests (e.g., ANOVA, t-tests) and their outcomes should be explicitly stated.

In this manuscript, authors generated flavonoids (naringenin) inducible gene expression system, which was transformed into a probiotic *E. Coli* strain to control functional gene expression. Both in vitro and in vivo results demonstrated the induced protein expression. Because the authors have published a very similar work for developing the naringenin-induced protein expression system, the in vitro work from this manuscript has no novelty. It is important to explore the feasibility of applications of a naringenin-responsive genetic circuit, but lots of defects were noticed.

1. They described that EcN colonized the mice gut, but there are no experiments performed, and no data verified its colonization.
2. Similar with the above, the EcN strain has chromosomal inserted *gfp* gene, but expression of GFP was not detected which may indicate the growth and colonization of this strain.
3. The in situ mice model used applied streptomycin treatment, I thought the purpose was to disturb the normal gut microbiome, but the authors did not discuss any details related to this like the limitations of this model for further translation applications.
4. Ampicillin was also used during the treatment to keep the plasmid, but at the same time ampicillin will also eliminate lots of gut microbes in the gut, making it not a real normal gut condition. As it has been done for inserting *gfp* gene into the chromosome, it is better to integrate the target gene to the genome of the probiotic strain.
5. No discuss about the dose use (100 mg/kg of naringenin) is comparable to that from the naringenin rich food?
6. No clear discussion about the weak peak came after the second induction compared to the first gavage treatment.
7. In Figure 2B, time is labeled as days, it is better to label the 2 h and 6 h because the results were described as hours.
8. In Table S1, some of the strains used/generated in this study were not listed.

Response to Reviewers

Dear Editor, Dr Ilana Kolodkin-Gal,

We appreciate the opportunity to revise our manuscript and are grateful for the constructive feedback provided by Reviewer #1 and Reviewer #2. Below, we present our detailed responses, highlighting the changes made to address each point raised.

Reviewer #1 (Comments for the Author):

In this manuscript, authors generated flavonoids (naringenin) inducible gene expression system, which was transformed into a probiotic *E. Coli* strain to control functional gene expression. Both in vitro and in vivo results demonstrated the induced protein expression. Because the authors have published a very similar work for developing the naringenin-induced protein expression system, the in vitro work from this manuscript has no novelty. It is important to explore the feasibility of applications of a naringenin-responsive genetic circuit, but lots of defects were noticed.

1. They described that EcN colonized the mice gut, but there are no experiments performed, and no data verified its colonization.

In order to address this issue, we have incorporated Supplementary Figure S3, which illustrates host colonization by EcN/114-*fdeR-nanoluc*. This figure encompasses CFU counts derived from fecal samples. Furthermore, qPCR was utilized to normalize data at time points where CFU counts were deemed unreliable. The following text was amended in pages 5-6: “Plasmid quantitation by qPCR is preferred over CFU counting for luminescence normalization as it provides a more accurate and consistent measure of plasmid abundance, minimizing variability caused by differences in bacterial viability and extraction from fecal samples”. A recent reference which highlighted the use of qPCR for bacterial quantification in fecal samples was also cited.

2. Similar with the above, the EcN strain has chromosomal inserted gfp gene, but expression of GFP was not detected which may indicate the growth and colonization of this strain.

Supplementary Figure S3 now incorporates a photograph of fluorescent EcN colonies which grew onto a LB agar plate, thereby corroborating the expression of sfGFP and the presence of EcN in the gut.

3. The in situ mice model used applied streptomycin treatment, I thought the purpose was to disturb the normal gut microbiome, but the authors did not discuss any details related to this like the limitations of this model for further translation applications.

The manuscript has been amended to include the following paragraph in the Discussion section (page 9-10), which emphasizes the employment of antibiotic-treated mice to replicate germ-free-like conditions:

“In addition, we employed an antibiotic-treated mouse model to evaluate the functionality of a naringenin-responsive genetic circuit in the gut. While this model has limitations in terms of its direct applicability to human microbiota contexts, it serves as a valuable first step for demonstrating the potential of flavonoids as inducers of gene expression in vivo. By reducing the complexity of the gut environment, this approach allows for a clearer assessment of the genetic circuit's performance without interference from native microbial populations. The controlled conditions ensure that observed changes in luminescence can be attributed to the activation of the genetic circuit by naringenin, rather than confounding interactions with the existing microbiota.”

4. Ampicillin was also used during the treatment to keep the plasmid, but at the same time ampicillin will also eliminate lots of gut microbes in the gut, making it not a real normal gut condition. As it has been done for inserting gfp gene into the chromosome, it is better to integrate the target gene to the genome of the probiotic strain.

Although we acknowledge this limitation, the primary objective was to demonstrate the efficacy of naringenin in activating genetic circuits within complex environments. Subsequently, we included the following paragraph in the Discussion section:

“Besides, future work could explore alternative strategies for maintaining plasmid selection that do not rely on antibiotics. One promising approach is auxotrophic markers, where the engineered strain is complemented with a plasmid carrying a gene essential for survival in a defined medium. This strategy eliminates the selective pressure exerted by antibiotics and reduces the disruption of the natural gut microbiota, making the system more applicable to real-world scenarios. Additionally, auxotrophic selection can provide a more stable and environmentally

friendly method for plasmid maintenance, facilitating long-term studies and potential therapeutic applications in more natural microbiota contexts. Such efforts would enhance the translational potential of flavonoid-responsive genetic circuits for use in probiotics and other living therapeutics.”

5. No discussion about the dose use (100 mg/kg of naringenin) is comparable to that from the naringenin rich food?

The following paragraph was amended to the Discussion section:

A naringenin dosage of 100 mg/kg to mice translates to approximately 2 mg for a 20-g mouse. To contextualise this dosage regarding human dietary intake, it is important to consider the naringenin content in common citrus fruits, such as grapefruits, achieving naringenin concentrations ranging from 62 to 308 mg/kg. Considering these values, achieving a naringenin intake of 100 mg/kg in humans through diet alone would require consuming an impractically large quantity of citrus fruits. Therefore, while dietary sources can contribute to naringenin intake, they are unlikely to reach the concentrations used in our experimental setup. For therapeutic applications aiming to replicate the effects observed in our study, naringenin supplementation or pharmacological formulations containing the EcN strain and an activator dose of naringenin may be more practical and effective than relying solely on dietary intake.

6. No clear discussion about the weak peak came after the second induction compared to the first gavage treatment.

The following paragraph was amended to the Results section (Page 7) addressing this observation:

“The temporal expression pattern is also noteworthy, with a peak in the first 6 h and a decline to near-control levels after 24 h. This pattern repeated after the second NAR gavage, although peak luminescence values were lower than those observed after the first dose. Likely, the emergence of an EcN population with reduced or silenced *nanoluc* expression has contributed to the luminescence reduction after the second gavage with naringenin, a possibility that needs to be addressed in further works.

7. In Figure 2B, time is labeled as days, it is better to label the 2 h and 6 h because the results were described as hours.

As suggested, Figure 2B has been updated to explicitly label the 2 h and 6 h time points.

8. In Table S1, some of the strains used/generated in this study were not listed

Table S1 has been updated to include all strains used or generated in this study.

Reviewer #2 (Comments for the Author):

The study entitled "Harnessing flavonoids to control probiotic function: in situ application of a naringenin-responsive genetic circuit" lays a strong foundation for the use of flavonoid-responsive genetic circuits in probiotic applications. However, challenges related to dose optimization, solubility, and interindividual variability must be addressed to unlock the full potential of this system. Future research should focus on integrating these circuits into more complex therapeutic strategies, including combination therapies and feedback-controlled systems, to create next-generation living medicines. The authors could enhance the clarity and impact of their work, following the below suggestions.

1. Abstract

Revise the abstract to remove errors, avoid redundancy, and streamline the presentation of results and conclusions. e.g. repeated phrases such as "as it presented minimal levels of basal output" occur, which disrupts the flow of information. Typos such as "probiotics" and "timetime". The authors are encouraged to conduct thorough proofreading to eliminate typographical errors, awkward phrasing, and formatting issues.

The abstract has been thoroughly revised to remove redundancies, streamline information, and enhance readability. Typographical errors and awkward phrasing have been corrected.

2. Importance

*** The introduction spends excessive space reiterating basic concepts about probiotics, synthetic biology, and flavonoids, which may be redundant for the target audience of a specialized publication. Example, "certain microbes are probiotics, offering health benefits**

to their hosts when taken in appropriate amounts" is unnecessary for readers familiar with the field.

*** The introduction does not sufficiently highlight the knowledge gaps addressed by the study, such as the challenges of achieving consistent genetic circuit activation in complex gut environments. Example, a more direct comparison between the performance of genetic circuits in vitro and in situ could frame the novelty of the work more clearly.**

*** Expand the discussion on why flavonoids are particularly promising inputs by addressing their gut stability, metabolic pathways, and interactions with microbial systems.**

*** Balance the level of detail across sections to create a cohesive narrative. For instance, provide a brief explanation of why E. coli Nissle 1917 was chosen, alongside details about the FdeR-PfdeA system.**

The Introduction has been systematically restructured to mitigate redundancies and highlight the innovative aspects of our work. We now distinctly articulate the existing knowledge gaps, including the challenges associated with maintaining consistent genetic circuit activation in vivo, and present a well-founded rationale for the selection of E. coli Nissle 1917 and the FdeR-PfdeA system.

3. Materials & methods

*** Streamline the text by eliminating repeated phrases and consolidating similar information. For instance, combine the fluorescence and luminescence measurement descriptions into one cohesive statement.**

*** Explain why GFP and Nanoluc reporters were chosen and how their use contributes to the study's goals. For instance, discuss the sensitivity or stability advantages of these reporters in gut environments.**

*** Include specific details such as the naringenin concentration range, plate reader settings, and qPCR cycling conditions in the main text to enhance reproducibility.**

*** Add critical details about the animal model, including the number of mice, grouping strategies, any pre-treatments like antibiotics, and ethical compliance.**

*** While supplementary material can contain extensive protocols, ensure that the main text provides sufficient standalone information for readers to grasp the experimental setup.**

Repetitive descriptions pertaining to fluorescence and luminescence measurements have been systematically consolidated. Additional details, including the range of naringenin concentrations, plate reader settings, and qPCR conditions, have been incorporated. Furthermore, information regarding animal models, pre-treatments, and ethical compliance is now clearly articulated in the Methods section

4. Results

*** While the study uses 100 μ M naringenin and 100 mg/kg doses, it does not explore the full dose-response relationship to assess threshold levels or optimize induction conditions.**

Example, no mention is made of how lower or higher doses influence the circuit, limiting the understanding of its sensitivity and range of application.

*** While the temporal decline in gene expression is acknowledged, the study does not explore how repeated dosing or alternative administration strategies could sustain circuit activity. Example, administering naringenin via food or developing water-soluble derivatives might overcome the solubility issue and ensure prolonged activation.**

*** The results rely on luminescence normalized to plasmid DNA concentration, but there is no discussion of variability in plasmid retention or expression across the bacterial population in the gut. Example, plasmid stability under gut conditions could vary, potentially influencing circuit performance.**

*** Although differences between groups are described as significant, the methods for determining significance are not detailed, leaving readers uncertain about the robustness of these claims. Example, specific statistical tests (e.g., ANOVA, t-tests) and their outcomes should be explicitly stated.**

An additional paragraph has been incorporated into the Results section (page 7), addressing the potential variability in plasmid retention and its effects on luminescence normalization.

The statistical methodologies have been clearly articulated in both the Methods section and the legend of Figure 2. We utilized the Mann-Whitney test, with significance threshold established at $P < 0.05$, to ensure clarity and transparency in our analysis.

Re: Spectrum02890-24R1 (Harnessing Flavonoids to Control Probiotic Function: In Situ Application of a Naringenin-Responsive Genetic Circuit)

Dear Prof. Marcelo Müller-Santos:

I enjoyed reading your revised manuscript. I appreciate the accurate responses to the reviewers and the candid discussion regarding the limitations of plasmid selection that depends on antibiotics.

Your manuscript has been accepted, and I am forwarding it to the ASM production staff for publication. Your paper will first be checked to make sure all elements meet the technical requirements. ASM staff will contact you if anything needs to be revised before copyediting and production can begin. Otherwise, you will be notified when your proofs are ready to be viewed.

Sincerely,
Ilana Kolodkin-Gal
Editor
Microbiology Spectrum